# Dating Violence among Undergraduate Medical Students at a Public University in Mexico City: An Exploratory Study

**DOI:** 10.3390/ijerph20043104

**Published:** 2023-02-10

**Authors:** Claudia Díaz Olavarrieta, Antonio Rafael Villa, Benjamin Guerrero López, Ingrid Vargas Huicochea, Sandra García-Medina, Monica Aburto Arciniega, María Alonso Catalán, Germán E. Fajardo Dolci, Ma. Elena Medina-Mora Icaza

**Affiliations:** 1Department of Psychiatry and Mental Health, Faculty of Medicine, National Autonomous University of Mexico, 3000 Ave. Universidad, Copilco Universidad, Coyoacán, Ciudad Universitaria, Mexico City 04510, Mexico; 2Research Division, Faculty of Medicine, National Autonomous University of Mexico, 3000 Ave. Universidad, Copilco Universidad, Coyoacán, Ciudad Universitaria, Mexico City 04510, Mexico; 3National School of Biological Sciences, National Polytechnic Institute, Prolongación de Carpio y, Plan de Ayala Street, Santo Tomás, Miguel Hidalgo, Mexico City 11340, Mexico; 4Faculty of Medicine, National Autonomous University of Mexico, 3000 Ave. Universidad, Copilco Universidad, Coyoacán, Ciudad Universitaria, Mexico City 04510, Mexico; 5Faculty of Psychology, National Autonomous University of Mexico, 3000 Circuito Ciudad Universitaria, Mexico City 04510, Mexico; 6National Institute of Psychiatry, Ramón de la Fuente Muñiz, Mexico City 14370, Mexico

**Keywords:** dating violence, gender-based violence, Mexico, medical students

## Abstract

Gender-based violence (GBV) and cyber-aggression are growing problems in Mexico, but there is a dearth of information on their associated risks. We aimed to determine the prevalence of dating violence (DV) and cyber-aggression in a public campus and compared students’ acceptability of abusive DV based on their sex and sexual orientation. We employed a cross-sectional design to survey 964 first-year medical students attending a public university. We analyzed who found “acceptable” abusive behaviors from a dating partner and carried out descriptive analyses of sample characteristics by sex. We included 633 women and 331 men. Homosexual and bisexual orientation was lower among women (1.5%, 4.8%) vs. men (16.9%, 7.2%). Of women and men, respectively, 64.2% and 35.8% reported having been in a dating relationship. Experiencing abusive behaviors in the year prior to the study was associated with students’ level of “acceptability”. A total of 43.5% of the students who experienced cyber-aggression did not report any mental health consequences, 32.6% did not seek professional help, and 17.4% reported feeling depressed. Students that accepted emotionally abusive DV behaviors displayed a fourfold risk of experiencing physical abuse. Women and sexual minorities are more at risk of experiencing GBV and DV. More male students reported being victims of cyber-aggression.

## 1. Introduction

In Mexico, gender-based violence (GBV) is a public health concern and a crime that includes “physical, sexual, emotional and financial abuse.” [1]. In 2007, the National Youth Institute reported that 15% of participants aged 15–24 had been victims of physical violence and 16.5% were victims of sexual violence [2]. A national survey from 2016 (ENDIREH) [3] reported that, of the country’s total population of 46.5 million, among women 15 years and older, 66.1% (30.7 million) had experienced some form of violence (psychological, physical, sexual, or financial) or workplace discrimination at least once in their lives. The survey also reported a prevalence of sexual violence of 41.3% during the 12 months prior to data collection; 45 per 100 women had experienced psychological, sexual violence, and workplace discrimination [3]. 

Dating violence (DV) can be understood within the context of social learning theories [4], a feminist perspective [5], and Bowlby’s attachment theory [6]. A 2007 Mexican national survey on dating violence [2] among 15–24-year-old participants reported a 39.38% prevalence of emotional abuse, 6.79% of physical abuse, and 8.16% for sexual abuse. Other researchers have explored the associations between DV and female empowerment among Mexican students [7], access to support networks and experiences of victimization [8], and personality types (using the Myers–Briggs inventory), [9] DV, defined as a form of “unperceived” violence among Mexican students [10], has myriad adverse effects on the well-being of young people and has been associated with risk factors, including family background and peer influence [11,12]. While the impact of DV has been documented [11,13], there is insufficient information on the long-term effects on academic performance [14,15]. The university climate may also result in a predisposition to perpetrate and experience DV among students [16], as this is strongly influenced by prevailing attitudes and perceptions during adolescence [17], especially in contexts such as Mexico where male predominance is pervasive, women’s compliance and acceptance of controlling and abusive behaviors are normalized [18] and all within the context of a highly homophobic society [19,20,21]. 

Cyber-aggression or cyberbullying is an emerging problem globally. This social problem was recently typified as a crime under a 2019 Mexico City law called the Olympia Law [22]. It is defined as intentional harm delivered by electronic means to a person or a group of people, irrespective of their age, who perceive(s) such acts as offensive, derogatory, harmful, or unwanted [23,24]. Social media, text messaging, and a wide range of online tools are used to practice sexual cyber-aggression, whose contents can include verbal or visual information about “sexual relationships, courtship, or sexual acts” [25] that may be explicit or non-sexually explicit [26,27]. Alcohol, Ref. [28] drugs, a hookup culture [16], and revenge porn also hinder our ability to measure the effects of cyber-aggression or define its boundaries [29].

Sexual and gender minorities are more likely to be victims of physical and sexual violence than the general population [30]. Violence against individuals based on their sexual orientation is another way in which sexual stigma can be expressed. In some countries, laws criminalize sexual and gender minorities directly or indirectly on the grounds of morality or the promotion of non-traditional values. This can result in physical punishment, the death penalty, arbitrary arrest and torture, ill-treatment in health facilities, and forced sterilization. Homophobic and transphobic violence may be physical or psychological, and it constitutes a form of GBV aimed at punishing those seen as defying gender norms [31,32]. 

Most of the data on the risks associated with DV, gender-based violence (GBV), campus-based sexual harassment (SH), and cyberbullying come from industrialized countries; there is a dearth of information from middle-income countries [33] where there is a significant need to create “safe spaces” within university settings to protect and support students. Due to Mexico’s current epidemic of GBV and femicides [34], we deemed it urgent to explore this pressing issue among undergraduate students at the largest public university campus in Latin America, which has approximately 229,268 students [35] spread throughout several campuses within Mexico City’s metropolitan area. At the Faculty of Medicine, women comprise over two-thirds of the student body [35], and thus it is important to document their DV experiences and the prevalence of cyber-aggression systematically. 

Measuring DV and campus-based sexual harassment has become more reliable thanks to robust data collection methodologies, such as campus climate surveys [16]. Exploring the magnitude of this widespread problem can inform interventions and raise awareness among students about its mental health consequences and academic performance. A systematic review of ethical and methodological issues in DV research in Mexico presses for the need to foster heterogeneity and conceptual definitions that will allow for greater consensus in the field [36]. With this in mind, we designed this study to determine the prevalence of dating violence (physical and sexual abuse), cyber-aggression (defined as unsolicited or non-consensual sexting), and bullying on social media (by a dating or romantic partner), which are all forms of gender-based violence. We explored their association with students’ acceptability of certain emotionally abusive dating behaviors (e.g., arguing often, insulting, demeaning, and threatening his/her partner). While gender plays an unequivocal role in the risk of experiencing GBV and DV, we wanted to explore if sexual orientation and/or gender in this student population of a country that may have invented machismo is a more significant risk factor for experiencing sexual and physical abuse [37] and cyber-aggression. As future healthcare providers, we were also interested in documenting how they are being socialized into conforming or non-conforming and homophobic attitudes vis a vis this social epidemic that has reached college campuses in Mexico [7,8,9,20,38,39,40]. 

## 2. Materials and Methods

### 2.1. Aim

The aim of this study was to analyze DV and cyber-aggression experiences among medical students based on their gender and sexual orientation.

### 2.2. Setting 

We employed a cross-sectional design. In April 2018, we surveyed first-year medical students attending the main campus at the largest public university in Mexico City (estimated population: 22 million). These students are exposed to GBV and other safety issues associated with living in a megacity [41]. Participants responded to an online self-administered survey upon agreeing to participate via written informed consent (IC). The survey included demographic data and 6 questions, and it took approximately 8 minutes to respond.

### 2.3. Sample 

Entry-level medical students were invited to participate after completing their first year. They were asked about their experiences with physical and sexual abuse and cyber-aggression (unsolicited sexting or bullying on social media) by a dating and or romantic partner in the past 12 months. As part of demographic data, we asked students about their sexual orientation and sexual identity as homophobic incidents on campus are increasing and the LGBT community has made a call to action [42]. We used the questions included in the 2018 survey on discrimination due to sexual and gender identity [43,44,45]: (a) Which of the following best describes your sexual orientation? Homosexual (lesbian or gay) (emotional attraction towards people of the same gender), bisexual (emotional erotic attraction to people of the same gender and opposite gender), heterosexual (emotional erotic attraction to people of the opposite gender), or other. (b) What is your gender identity? woman, man, trans woman (transsexual, transgender, transvestite), trans man (transsexual, transgender, transvestite), or other. Unlike the United States or Canada, where students complete pre-medical courses before being admitted into medical school, in Mexico and regionally, students apply once they graduate from high school. Thus, the mean age when we first approached participants to explain the nature of the study and invite them to answer the survey was relatively young (18.8 years). 

### 2.4. Study Instrument

The survey was adapted from questionnaires previously administered in Mexico [46]. We performed an internal reliability analysis using Cronbach’s Alpha (0.88) and included questions on students’ experiences and acceptability of emotionally abusive DV behaviors. We measured the prevalence of physical and sexual abuse and cyber-aggression experiences (past 12 months). Cyber-aggression was defined as having had experiences of unsolicited sexting and bullying on social media by a dating partner (past 12 months). In January 2018, we carried out a pilot test to validate the survey among students with similar characteristics as our sample (n = 100). After concluding the pilot phase, we made the necessary adjustments and adaptations to the study instrument.

### 2.5. Sample Size

All first-year medical students (n = 1364) were invited to participate; 964 students finally took part in the electronic survey, and their mean age was 18.8 years. Of the 964, 64.8% (n = 625) reported being in a dating relationship in the last 12 months. Inclusion criteria: Students enrolled in the class of 2019, attending the main campus, who had granted written IC to participate. Exclusion criteria: Students who did not give their IC. Sampling: Due to the sensitive nature of the study and because students were volunteers, we used convenience non-probabilistic sampling. We included the students’ parents’ level of schooling as a proxy for socioeconomic status [35]. We wanted to explore if the student’s family background placed them at a greater social disadvantage and increased their risk of experiencing GBV and DV [47,48,49]. 

Our dependent variables included DV experiences (past 12 months): physical and sexual abuse and cyber-aggression that included students who responded affirmatively to one of the following questions: Physical abuse included students who responded affirmatively to the question (did someone you dated, harm or injure you?); sexual abuse included students who responded affirmatively to the question (did someone you were dating forced you to do sexual things against your will?. Please include kissing, touching, or having been physically forced to have sexual intercourse against your will); and for cyber-aggression the questions included: did someone you dated, post negative/offensive comments on social media, did someone you dated show private or intimate photos of you, and how many times were you a victim of sexting, bullying or cyber-aggression? 

Independent variables included students who reported “acceptable” emotionally abusive behaviors from a dating partner. These included: (i) arguing frequently, (ii) insulting or belittling my partner, (iii) controlling my partner schedules or daily routines, (iv) preventing my partner from seeing her or his family, (v) threatening my partner verbally, (vi) not allowing my partner to work or study, (vii) tell my partner what she can or cannot do, (viii) control my partner finances, (ix) control the type of contraceptive method and (xii) prevent my partner from using any contraceptive method. We also enquired about the mental health consequences and help-seeking behaviors among victims of cyber-aggression and included a dummy variable indicating whether students responded affirmatively to accepting at least one of these behaviors (argue often; insult or despise your partner, etc.), which allowed us to identify students who accepted emotionally abusive behaviors from their dating partners that we construed as DV. 

Students participated voluntarily and signed a written IC. An electronic link to a factsheet that included information on campus resources on GBV and DV was available. Each participant took a copy of their IC, and the research team kept a copy of the study files. The IC included the university Research Division’s criteria for confidentiality, storage, and data analysis. Students did not receive any financial or academic incentive for their participation. The study was approved by the Faculty of Medicine Research and Ethics Commission (IRB Number: FM-DI-088-2018). 

### 2.6. Statistical Analysis

We carried out descriptive analyses of sample characteristics by sex. We calculated prevalence estimates for individual experiences of physical abuse, sexual abuse, and cyber-aggression by a dating partner expressed as the percentage of students who answered affirmatively to each one of the questions described above. We first compared the distribution of the proportion of each of the sample characteristics (gender and parents’ level of schooling) with each one of the DV experiences (physical, sexual abuse, and cyber-aggression) using Fisher’s exact test (bilateral significance). We measured the students’ level of "acceptability" or agreement with emotionally abusive DV behaviors with our dependent variables: physical abuse, sexual abuse, and cyber-aggression, using Fisher’s exact test (bilateral significance) and a linear trend test. We performed bivariate analysis to identify the association between sex, "acceptability" of emotionally abusive DV behaviors, and the following dependent variables: physical abuse, sexual abuse, and cyber-aggression. We carried out logistic regression models that included variables that, in the bivariate analysis, had a value of *p* < 0.20. We calculated odds ratios (OR) with a 95% confidence interval (95% CI). Significance levels were set at 0.05, and the analysis was performed with SPSS (SPSS/PC v23.0) [50].

## 3. Results

A total of 964 students agreed to participate; 2/3 were women (n = 633). The mean age was 18.8 (SD = 1.0). Homosexual and bisexual orientation, by self-report, was lower among women (1.5%, 4.8%) vs. men (16.9%, 7.2%), respectively. Of all students, 64.2% of women (n = 401) and 35.8% of men (n = 224) reported having been in a dating relationship in the last 12 months. Table 1 presents the characteristics of the study participants. The prevalence of physical and sexual abuse and cyber-aggression is included in Table 2. Experiencing cyber-aggression, physical abuse, and sexual abuse in the year prior to the study were associated with students’ acceptability of DV behaviors. The association between cyber-aggression, bullying, and physical abuse with an acceptance of DV behaviors yielded statistically significant results (see Table 3). 

At least one positive response to one of the following behaviors: (i) arguing frequently, (ii) insulting or belittling my partner, (iii) controlling my partner schedules or daily routines, (iv) preventing my partner from seeing her or his family, (v) threatening my partner verbally, (vi) pushing or hitting my partner when you get angry, (vii) not allowing my partner to work or study, (viii) tell my partner what she or he can or cannot do, (ix) control my partner finances, (x) force my partner to have sex against her or his will, (xi) control the type of contraceptive method, and (xii) prevent my partner from using any contraceptive method.

We asked students if they considered emotionally abusive DV behaviors as “acceptable” to gauge how these adhere to cultural standards of normalization of violence prevalent in Mexican society [51]. Arguing often was the only behavior that over half (60.4%) of male students found acceptable compared with female students, which yielded significant results (0.001). Gender was not a variable that was associated with experiencing physical and sexual abuse but accepting DV behaviors and experiencing cyber-aggression was (in the adjusted and the non-adjusted models). Accepting DV behaviors had an OR of 6.3 (*p* = 0.02), experiences of physical abuse were adjusted by gender and cyber-aggression; experiences of sexual abuse had a borderline risk (or = 2.3, *p* = 0.07) adjusted with the same variables. Cyber-aggression experiences were associated with having experienced physical (adjusted OR 4.2, *p* = 0.01) and sexual abuse (adjusted OR = 3.5), *p* = 0.02 (see Table 4) by a dating partner. Being a victim of cyber-aggression (irrespective of sex) seems to place students at more risk of experiencing other forms of GBV, such as physical and sexual abuse.

While the mental health consequences of cyber-aggression have been documented extensively [52,53], this question was left unanswered by most participants. Almost half (43.5%) of those who reported experiences of cyber-aggression (unsolicited sexting and bullying on social media) did not report any mental health consequences, 32.6% did not seek professional help, and 17.4% reported feeling depressed (17.4% males, 12.8% females). We need to explore in depth if this reflects the students’ perceptions of the stigma associated with mental health or if the normalization of web-based violence and hate speech have reached a level of saturation among young people [54]. We carried out a logistic regression analysis to calculate the odds of experiencing physical and sexual abuse by a dating partner (last 12 months) among those who reported being in a relationship (n = 625). Notably, gender was not associated with the risk of experiencing sexual or physical abuse. Students that accepted emotionally abusive DV behaviors (argue often; insult or despise your partner, etc.), displayed an increased risk of experiencing physical abuse both in the non-adjusted and the adjusted estimates. These behaviors displayed a borderline association with sexual abuse in the non-adjusted and adjusted values (see Table 5). Experiencing cyber-aggression was also associated with a higher probability of experiencing both physical and sexual abuse in the adjusted and non-adjusted estimations (see Table 5).

## 4. Discussion

The aim of the present exploratory study was to document medical students’ experiences of physical and sexual abuse and cyber-aggression and their acceptability of emotionally abusive DV behaviors at the end of their first year of undergraduate training at a public university. To our knowledge, this is the first attempt at exploring personal experiences with DV and cyber-aggression on a public college campus in Mexico City. Following Shorey’s description of theoretical frameworks to help explain dating violence [6], our results can be interpreted under Bandura’s social learning theory (behaviors are learned through observation and imitation of other’s behavior and maintained by reinforcement), especially within the traditional context, and a rigid patriarchal societal system (feminist theory) and follows Joly LE and Connolly J’s [55] definition of high-risk young women who may be prone to experiencing DV. However, some instances of dating violence could be explained under Bowlby’s attachment theory, where adolescents follow family prototypes as templates for their initial romantic attachments [6]. Our student sample reflects a larger context; the campus is in Mexico City, the world’s femicide capital [56], and our results are consistent with the interaction that has been described among private, state, and structural violence [57]. In response to sexual harassment incidents on the main campus, the university, together with the student’s ombudsman [58] and campus police, has become very active in addressing this ongoing problem that, for some researchers, needs to be addressed with primary prevention strategies and initiated earlier, when dating violence first becomes a public health concern. Strategies involving a multi-level approach (school, home, and community) [59] that not only involve curricular content but help students recognize warning signs of DV to include attention to gender equity, gender identity, sexual attraction, and sexual and reproductive health education may have a greater impact. Other interventions that include parental monitoring (parents knowing about child´s activities, children disclosing these activities, and rule setting) and the involvement of families also appear promising (i.e., Families for Safe Dates, Project STRONG, and components of the Centers for Disease Control and Prevention multicomponent program “Dating Matters”) [60,61,62] compared in an RCT (Dating Matters vs. Safe Dates) with middle school students showed, on average, lower DV rates of perpetration and victimization and lower use of negative conflict resolution strategies [63].

Widespread media campaigns (in collaboration with the city government) have also lent visibility to GBV and SH. The university is also working with the Mexico City subway system to help disseminate prevention media campaigns [64].

The prevalence of physical abuse and sexual abuse among male and female students by a dating partner was 3.0 and 6.1%, respectively. Our figures are comparable to, albeit smaller than, Mexico´s national dating survey of 6.79% for physical abuse and 8.16% for sexual abuse. Our inclusion criteria to define a dating or romantic partner was different and reminds us of the need to be consistent in conceptual definitions of DV following Rojas Solis et al. [36]. Gender was not considered a risk factor for experiencing sexual or physical abuse; however, more male students self-identified as gay (4.8% vs. 1.5%) or bisexual (16.9 vs. 7.2%) compared to women. This prevalence is almost a fivefold increase in the last ten years compared with a local, national survey report that sampled 25,630 adolescents (men and women) aged 12–19 [65] and yielded the following prevalences: 0.9% for homosexual; 0.7% bisexual, and 0.1% lesbian identity. Our finding is consistent with data from the United States that also shows an increase over time in how respondents self-identify; in 2012, 3.4% of male respondents identified as LGBT compared with 4.9% who did so in 2020 [66]. Furthermore, a recent WHO study documented the association of physical and sexual violence with the perception of sexual orientation and gender identity and found this problem is compounded in societies such as Mexico’s, where the status of women is low and femininity and homosexuality are denigrated and discredited [32]. It is plausible to assume that perhaps young medical students at the university who self-identify as gay or bisexual may be more at risk of experiencing physical or sexual abuse. Gender did play a role in cyber-aggression; more male students reported being victims of bullying and unsolicited sexting, a form of aggressive behavior. However, almost two-thirds of participants (71.7%) were unable to confirm the perpetrator’s gender.

Sexual violence implies a power differential [67], and our study’s prevalence for sexual abuse (by a dating partner) was 6.1%, a number that seems to be lower compared with other countries where sexual assault is perhaps reported more consistently [68,69]. However, we surveyed first-year medical students, and recent data suggest women’s risk of victimization increases over a four-year period (to an incidence of 20–25%) and thus will be lower during their first two years of college [70] where an increase in partying and alcohol make students more vulnerable to sexual abuse. The university has approximately 229,268 students [31]. If we extrapolate our 6.2% conservative prevalence to all campuses, this translates into 13,985 undergraduates who, by self-report, could disclose being victims of sexual abuse by someone they dated in the past 12 months. However, findings from the United States and Canada seem to indicate the onus of prevention strategies should not be approached on isolation (focusing on victims) but instead needs to address the structural inequities that promote and perpetuate sexual violence [71] and allow us to implement, in a middle-income country, a comprehensive multi-level approach as described by Senn [70].

Our study sample was young (mean age 18.8), yet according to the number of hours per day young people spend on average on social media [72], experiencing cyber-aggression by a dating partner increased their risk of being victims of physical and sexual abuse fourfold, which may point towards a continuum of different forms of abuse (see Table 5). Cyber-aggression through social media affects male and female students equally [73]; however, in our study, experiences of cyber-aggression were more often reported by male students. 

Compared to physical (3.0%) and sexual abuse (6.1%), the prevalence of cyber-aggression was the highest (7.1%). It is also worth noting that almost half of the participants (43.5%) said they did not experience any mental health consequences due to cyber-aggression, which may reflect the stigma associated with mental health or tolerance towards this new form of social interaction [74,75]. Gender in medicine continues to play a controversial role [76,77]; however, paradoxically, in this study, we did not find a difference between male and female students in their risk of experiencing physical or sexual abuse by a dating partner. While our results seem to contradict mainstream findings on this topic, Shorey cites evidence to document how females inflict psychological and physical aggression more often than males [6].

On 12 February 2020, the university’s president amended the guidelines to show zero tolerance towards GBV and SH in Mexico City and at state-level campuses. The university’s dissemination news outlet [78] announced the creation of a Commission on Gender Equity that will review and amend the curricula to include a gender perspective, one we hope will also further “raising the bar on sex and gender research reporting” [79]. Galende N. et al., using the PRISMA statement, carried out a systematic analysis to document universal cyber dating violence prevention programs to document their effectiveness and concluded that three programs fulfilled their search criteria. These were the DARSI program, the Dat-e Adolescence program, and the Brief Incremental Theory of Personality (ITP) adolescent dating violence prevention program. The Spanish research group also included a summary of the program contents covered in these prevention programs that included common content blocks: (1) helping a student develop socio-emotional competencies (e.g., self-esteem, empathy, emotional intelligence), (2) addressing gender stereotypes or sexist beliefs and attitudes (e.g., rigidity in the definitions of masculinity and femininity, gender inequality), and (3) addressing attitudes towards partner violence (e.g., beliefs that justify violence, types of abuse or roles in dating abuse). The authors conclude there is a dearth of information on middle-income or developing countries’ experience with rates of perpetration and victimization of this novel form of interaction among people [80]. The new Information and Communication Technologies (Zoom, Google Classroom) at universities, that before the SARS-CoV-2 pandemic served as optional learning tools, have now become the new vehicles for students to engage in anonymous, abusive behaviors as they spend more time in front of a screen and thus engage in cyber-aggression. We hope our study findings help improve students’ college experience, shed light on DV, and halt the spread of cyber-aggression. The restrictions imposed during confinement changed the way in which young people interacted with each other, including the use of online dating and sex, which may have increased the risk of cyber-aggression. Lastly, prevention programs that focus on youth, such as our study sample, facing marginalization (race and ethnicity, sexual orientation, or gender identity) deserve our utmost attention [60].

*Study limitations*: Data were collected by online assessment, which is not the optimal way of measuring sensitive behaviors. While our electronic survey was used due to cost, ease, and logistical constraints, online surveys have two significant methodological constraints: (1) they do not allow us to generalize our findings as we are unable to describe our population, and (2) self-selection bias among respondents may be over-represented [81,82,83,84]. The growing prevalence of online surveys also makes them more often reviewed by IRBs at colleges in the United States, thereby circumventing ethical issues in research with humans, including consent, risk, privacy, anonymity, confidentiality, and autonomy [85]. The consistency of data collected using online surveys has also been explored. A study with 5181 respondents found that a subset of participants asked to collect an online life history calendar showed similar recall among online respondents [86] compared with data that included four data collection methods: in-person, mail-in, Internet-based, and telephone surveys. Internet-based surveys had the highest response rate when collecting data on oral health when compared with Internet-based surveys (37%) [87]. Qualitative online data collection has also addressed some of its strengths and limitations. A quasi-experimental study on sensitive or dissenting information using focus groups and individual interviews reported participants’ perceptions on the collection process and found they felt less rapport and personal comfort when disclosing information on medical risk during pregnancy using the focus group modality when compared with other modalities such as in-person vs. online (video, online chat-based, and online email/message board-based) [88]. Other studies found that rapport, disclosure, and anonymity were features online methods needed to address; however, data equivalence seemed to point toward other issues (lower eligibility and completion rates and higher time and monetary costs for audio and video modes) [89]. The COVID-19 pandemic highlighted the need to explore alternative methods for data collection on sensitive topics. Saarijärvi M and Bratt EL [90] measured the impact of a transition program in Sweden for adolescents with congenital heart disease and concluded the gold standard (face-to-face interviews) may not always be cost-effective or promote inclusion diversity, and cost when comparing online vs. in-person recruitment have also shown that online surveys have the potential of reaching less healthy populations. However, they do not seem to be less costly when recruiting focus group participants [91].

Our cross-sectional methodology did not allow us to identify if the acceptance of emotional abuse during dating is a consequence of the context in which they live or if there is an interaction between these variables. We were unable to identify the point in time when students started accepting or agreeing with DV behaviors and did not enquire about the partner’s age or if the aggressor was campus-based. The prevalence estimates could be considered “low” because we only inquired about dating partners in the past 12 months. Future research should clarify if medical students are engaging in GBV and DV as victims and/or perpetrators with the aim of working to support prevention and response programs.

## 5. Conclusions

At this public university, more male students self-identified as gay compared with national adolescent sexual orientation surveys; this may have affected our physical and sexual abuse prevalence estimates. Mexico is a country where women and sexual minorities are more at risk of GBV and DV, which may help explain why gender, normally found as a risk for abuse, was not significant except for cyber-aggression experiences where more male students reported being victims of this form of abuse, although the gender of the perpetrator could not be confirmed. Our sample seems to point toward the way in which homophobia in social media, in a country like Mexico, seems to trigger cyber-aggression. It is a novel field of study, and we are only able to speculate on this finding. We also need to carry out further research with more sensitive methodologies that will allow us to ascertain the mental health consequences of cyber-aggression in the digital era and capture the subtle effects of the interactions young people are having on social media [92].

## Figures and Tables

**Table 1 ijerph-20-03104-t001:** Characteristics of study participants (n = 964).

	Women		Men		Differences between Genders	Total	
	$\bar{x}$	SD	$\bar{x}$	SD	*p* ^†^	$\bar{x}$	SD
Age (years)	18.8	1.0	18.9	1.3	0.05	18.8	1.1
	**n**	**%**	**n**	**%**	*p* ^§^	**n**	**%**
Who do you currently live with							
Both parents	413	65.2	200	60.4	0.01	613	63.6
Father	18	2.8	10	3.0		28	2.9
Mother	130	20.5	58	17.5		188	19.5
Alone/other/partner	72	11.4	63	19.0		135	14.0
Mother’s level of schooling *							
Less than high school	163	26.1	71	22.0	0.27	234	24.7
High school	225	36.0	131	40.6		356	37.6
College or more	237	37.9	121	37.5		358	37.8
Father’s level of schooling *							
Less than high school	135	23.2	77	24.5	0.82	212	23.6
High school	174	29.8	96	30.6		270	30.1
College or more	274	47.0	141	44.9		415	46.3
Sexual orientation (by self-report) *							
Homosexual	5	1.5	40	16.9	<0.0001	45	7.9
Bisexual	16	4.8	17	7.2		33	5.8
Heterosexual	313	93.7	179	75.8		492	86.3

^†^ *t*-test; ^§^ χ**²** test; * Missing values for these variables.

**Table 2 ijerph-20-03104-t002:** Prevalences of physical abuse, unsolicited sexting, bullying, sexual abuse, and cyber-aggression by gender and dating partner.

	Women		Men		Differences between Sex	Total	
	n	%	n	%		n	%
Did someone you were dating harm or injure you (last 12 months)							
Yes	10	2.5	9	4.0	0.33	19	3.0
No	391	97.5	215	96		606	97
Did you experience unsolicited sexting by a dating partner (last 12 months)							
Yes	16	4.0	15	6.7	0.18	31	5.0
No	385	96.0	209	93.3		594	95.0
Did you experience bullying (social media) by a dating partner (last 12 months)							
Yes	14	3.5	18	8.0	0.02	32	5.1
No	387	96.5	206	92.0		593	94.9
Did someone you were dating forced you to do sexual things against your will? * (last 12 months)							
Yes	20	5.0	18	8	0.16	38	6.1
No	381	95	206	92		587	93.9
Did you experience cyber/aggression ** by a dating partner (last 12 months)							
Yes	21	5.2	23	10.3	0.02	44	7.1
No	380	94.8	200	89.7		580	92.9

* (Please include kissing, touching, or physically forced you to have sexual intercourse against your will), ** unsolicited sexting + bullying.

**Table 3 ijerph-20-03104-t003:** Acceptability of DV *, cyber-aggression, sexual abuse, bullying, unsolicited sexting, and physical abuse; last 12 months (n = 625).

	DV Behaviors Were Considered						
	Acceptable		Unacceptable		p	Total Prevalences /625	
	n	%	n	%		n	%
Cyber-aggression (unsolicited sexting + bullying)	31	70.5	13	29.5	0.04	44	7.1
Sexual abuse	27	71.1	11	28.9	0.06	38	6.1
Bullying (social media)	24	75.0	8	25.0	0.03	32	5.1
Unsolicited sexting	21	67.7	10	32.3	0.20	31	5.0
Physical abuse	17	89.5	2	10.5	0.002	19	3.0

χ² test. * Acceptability of dating violence (DV) behaviors.

**Table 4 ijerph-20-03104-t004:** Association between sex, frequency of physical and sexual abuse, and cyber-aggression (by his/her dating partner).

	**Women** **633** **(65.7%)**		**Men** **331** **(34.3%)**		**Differences between Gender**	**Linear Trend**	**Total** **964** **(100%)**	
	**n**	**%**	**n**	**%**	** *p* **	***p* ***	**n**	**%**
How many times were you a victim of cyber-aggression by someone you dated? (unsolicited sexting + bullying)								
I have not dated or been in a dating relationship in the last 12 months	248	39.2	100	30.2	0.002	<0.0001	348	36.1
0 times	349	55.1	194	58.6			543	56.3
1 time	20	3.2	17	5.1			37	3.8
2+ times	16	2.5	20	6.0			36	3.7
How many times did someone you dated forced you to perform sexual acts?								
I have not dated or been in a dating relationship in the last 12 months	232	36.7	107	32.3	0.14	0.38	339	35.2
0 times	381	60.2	218	65.9			599	62.1
1 time	16	2.5	3	0.9			19	2.0
2+ times	4	0.6	3	0.9			7	0.7
How many times did the person you dated hurt or physically abused you?								
I have not dated or been in a dating relationship in the last 12 months	234	37.0	108	32.6	0.32	0.13	342	35.5
0 times	389	61.5	214	64.7			603	62.6
1 time	5	0.8	6	1.8			11	1.1
2+ times	5	0.8	3	0.9			8	0.8

χ² test; * χ² for linear trend.

**Table 5 ijerph-20-03104-t005:** * Univariate, multivariate logistic regression, physical, sexual abuse, acceptance of DV behaviors or experiencing cyber-aggression (n = 625).

Independent Variables	Experiences of Physical Abuse by a Dating Partner (Last 12 Months)				Experiences of Sexual Abuse by a Dating Partner (Last 12 Months)			
	Non Adjusted OR		Adjusted OR		Non-Adjusted OR		Adjusted OR	
	OR (95% CI)	*p*-Value	OR (95% CI)	*p*-Value	OR (95% CI)	*p*-Value	OR (95% CI)	*p*-Value
Gender								
Female	1.0		1.0		1.0		1.0	
Male	1.6 (0.7–4.1)	0.29	1.3 (0.5–3.2)	0.63	0.5 (0.2–1.3)	0.17	0.4 (0.2–1.1)	0.08
Acceptability of DV behaviors								
No	1.0		1.0		1.0		1.0	
Yes	7.2 (1.6–31.2)	0.009	6.3 (1.4–28.0)	0.02	2.3 (0.9–5.5)	0.07	2.3 (0.9–5.5)	0.07
Cyber-aggression experiences								
No	1.0		1.0		1.0		1.0	
Yes	5.2 (1.8–15.1)	0.003	4.2 (1.4–12.5)	0.01	3.4 (1.2–9.5)	0.02	3.5 (1–2–10.2)	0.02

* Non-adjusted and adjusted odds ratios physical and sexual abuse (1 = yes, 0 = n) according with acceptance of DV behaviors (1 = acceptable, 0 = unacceptable) or experiencing cyber-aggression among students in a dating relationship (past 12 months, n = 625).

## Data Availability

The datasets used and/or analyzed during the current study are available from the corresponding author upon reasonable request.

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
