# Peer review of "Dating Violence among Undergraduate Medical Students at a Public University in Mexico City: An Exploratory Study"

_ijerph, 2023, doi:10.3390/ijerph20043104_

Round 1

Reviewer 1 Report

In general, the manuscript is quite interesting. In the introduction, the relevant research topics are discussed. Bibliographical references are well used.

The biggest problem of this manuscript is that there isn't discussion.

The discussion should not re-present the results. The authors should explain the reasons for their results. I think the discussion must be improved.

Finally, the online assessment has serious limitations. I think they should be considered by the authors.

Reviewer 2 Report

It is always great to highlight findings, however, some recommendations are needed going forward. The paper is missing recommendations and some solutions to the issue. 

Reviewer 3 Report

The conclusions should be theoretically underpinned.

Reviewer 4 Report

This is an interesting and important contribution to the field of sexual violence prevalence studies, with a large sample of students, and a timely addition of cyber-aggression. It is well presented and the findings are explained well. However, there are some changes to be made to the discussion of the methods/results regarding ‘sexting’ - it is not specified whether this refers to unsolicited or non-consensual sexting, which would be an example of aggressive behaviour, or consensual sexting between dating partners, which would not be a form of aggressive behaviour. As this is reported by 5% of the students surveyed, it’s hard to know whether the reports of cyber aggression being higher than expected because this variable has been included. If this doesn’t explicitly refer to only non-consensual sexting, then I would propose that this variable be removed from the analysis. If it does only refer to non-consensual sexting, then this needs to be made much more explicit in the writing. 

There are some repeated issues with English which need correcting/discussing: 

  • - 96 Is the word 'feminicides' or femicides? 

  • - Sex and gender are sometimes used interchangeably throughout – but these aren’t interchangeable 

  • - There are some grammatical errors which sound odd to a native English speaker e.g.did someone with whom you went out on a date, harmed or injured you” would make more grammatical sense as “Did someone you dated, harm or injure you?”  

  • - Using 'her' when discussing acceptability of dating violence - although this is most commonly enacted against women, was this how the question was asked in the research? Quite a high percentage of male participants identified as gay, so how would they answer this? Or men who find dating violence acceptable when women perpetrate? 

There are some other minor areas to attend to: 

- 83-90 this section needs to be clarified. How are state violence and interpersonal violence related (if they are) 

- 116-117 You haven't really demonstrated in the Introduction that there is an epidemic of homophobia in universities? 

- Final paragraph of Intro - it seems as though this study is looking at lots of different aspects of GBV which need to be more clearly linked together in your Introduction. 

- 138 - Is homosexuality defined outside of erotic attractive? Emotional and affective attraction sound the same to me 

- 256-7 How was the relationship between cyber aggression and experience of sexual abuse investigated - are is CA actually predictive of SA or are they just commonly both experienced? 

- 326-331 Needs to be more discussion of why you think the experience of cyber aggression is higher for male students. This is contrary to expectations and other studies, so more attention needed to this issue.

Round 2

Reviewer 1 Report

The discussion of the results must be supported by the references used in the introduction. In this research, only three references from the introduction are used. Consequently, I believe that the authors have to improve the discussion and support their results with theoretical research.
